# Coding for Large-Scale Distributed Machine Learning

**DOI:** 10.3390/e24091284

**Published:** 2022-09-12

**Authors:** Ming Xiao, Mikael Skoglund

**Affiliations:** Division of Information Science and Engineering, Royal Institute of Technology, Malvinas Vag 10, KTH, 100-44 Stockholm, Sweden

**Keywords:** error-control coding, gradient coding, random codes, ADMM

## Abstract

This article aims to give a comprehensive and rigorous review of the principles and recent development of coding for large-scale distributed machine learning (DML). With increasing data volumes and the pervasive deployment of sensors and computing machines, machine learning has become more distributed. Moreover, the involved computing nodes and data volumes for learning tasks have also increased significantly. For large-scale distributed learning systems, significant challenges have appeared in terms of delay, errors, efficiency, etc. To address the problems, various error-control or performance-boosting schemes have been proposed recently for different aspects, such as the duplication of computing nodes. More recently, error-control coding has been investigated for DML to improve reliability and efficiency. The benefits of coding for DML include high-efficiency, low complexity, etc. Despite the benefits and recent progress, however, there is still a lack of comprehensive survey on this topic, especially for large-scale learning. This paper seeks to introduce the theories and algorithms of coding for DML. For primal-based DML schemes, we first discuss the gradient coding with the optimal code distance. Then, we introduce random coding for gradient-based DML. For primal–dual-based DML, i.e., ADMM (alternating direction method of multipliers), we propose a separate coding method for two steps of distributed optimization. Then coding schemes for different steps are discussed. Finally, a few potential directions for future works are also given.

## 1. Background and Motivations

With the fast development of computing and communication technologies, and emerging data-driven applications, e.g., IoT (Internet of Things), social network analysis, smart grids and vehicular networks, the volume of data for various intelligent systems with machine learning has increased explosively along with the number of involved computing nodes [1], i.e., in a large scale. For instance, learning systems based on MAPReduce [2] have been widely used and may often reach the data volume of petabytes (10${}^{15}$ bytes), which may be produced and stored in thousands of separated nodes [3,4]. Large-scale machine learning is pervasive in our societies and industries. Meanwhile, it is inefficient (sometimes even infeasible) to transmit all data to a central node for analysis. For the reason, distributed machine learning (DML), which stores and processes all or parts of data in different nodes, has attracted significant research interests and applications [1,3,4,5,6,7,8,9,10,11,12,13,14,15,16]. There are different methods of implementing DML, i.e., primal method (e.g., distributed gradient descend [4,7], federated learning [5,6]) and primal–dual method (e.g., alternating direction method of multipliers (ADMM)) [16]. In a DML system, participating nodes (i.e., agents or workers) normally process local data and send the learning model information to other nodes for consensus. For instance, in a typical federated learning system [5,6], worker nodes run multiple rounds of gradient descends (local epoch) with local data and received global models. Then, the updated local models are sent to the server for aggregating into new global models (normally weighted sum). The models are normally much shorter than raw data. Thus, significant communication costs are saved by federated learning, and meanwhile the transmission of models in general has better privacy than sending raw data over networks. Actually, in addition to federated learning, other DML also has the benefits of communication efficiency and improved privacy since model information has, in general, smaller volumes and better privacy than raw data.

Despite various benefits, there are severe challenges for the implementation of DML, especially for large-scale DML. Ideally, DML algorithms have speedup gains, which should scale linearly with the number of participating learning machines (computing nodes). However, the practical speedup gain of DML is limited by various bottlenecks, and is still far from the theoretical upper limits [17,18]. Among others, significant bottlenecks include communication loads, security, global convergence, synchronization, slow computing nodes, complex optimization functions, etc. For instance, due to the limitation of computing capability and communication networks, a part of the computing nodes may have slow response and become the bottleneck of DML systems if the fast-response nodes have to wait for them. These nodes are often referred to as straggler nodes [4], and also called system noise [19]. To efficiently combat the straggler nodes, many schemes have been proposed, such as repetition nodes [20,21], blacklisting straggler nodes [22] and error-control codes [4,8,9,10,11,12,13,14,23,24,25]. Blacklisting method detects the straggler nodes and will not schedule more tasks to them. Thus, it is a type of *after-event* approach. The repetition of computing nodes needs lots of resources and a suitable mechanism to detect straggler nodes and find corresponding repetition nodes. Moreover, it is rather expensive to repeat all computing tasks and related data. More recently, error-control coding was proposed for DML by regarding straggler nodes as erasure, which can be corrected by coded data from non-straggler nodes and are shown to be much more efficient than the schemes based on replication. Error-control coding can correct the loss by straggler nodes of current learning rounds and thus is a type of *current-event* approach.

In [8], more practical computing networks with hierarchical structures were studied. For such networks, hierarchical coding schemes based on multiple MDS codes were proposed to reduce computation time. In [9], each multiplication matrix was further divided into sub-matrices, and all sub-matrices were encoded by MDS codes (e.g., Reed–Solomon codes). Thus, the computed parts in straggler nodes can be exploited, and the computing time can be further reduced. However, as the number of nodes and sub-matrices increases, the complexity of the MDS codes will increase substantially. In [25], the deterministic construction of Reed–Solomon codes was proposed for gradient-based DML. The generator matrix of the codes in [25] is sparse and well balanced, and thus the waiting time is reduced for gradient computation. In [10], a new entangled polynomial coding scheme was proposed to minimize the recover threshold of master–worker networks with generalized configurations for matrix-multiplication-based DML. In [26,27], coding schemes are considered for matrix multiplication in heterogeneous computing networks. However, the complexity of coding in [26,27] is still very high for large-scale DML since matrix inversion is used for decoding, and moreover, the coding matrix is pre-fixed and is hard to adapt to varying networks. In [28], low-complexity decoding was proposed for matrix multiplication for DML. However, the results in [28] are preliminary and hard to be used for heterogeneous networks, and the communication load is still very high. In [11], coding schemes based on the Lagrange polynomial are proposed to encode blocks among worker nodes. The proposed codes may achieve optimal tradeoffs among redundancy (against straggler nodes), security (against Byzantine modification) and privacy. However, the coding scheme in [11] is also based on MDS codes, which may not be flexible and have high complexity for large-scale DML. Furthermore, the existing coding schemes are mostly for matrix multiplication (for distributed gradient descend), i.e., the primal method. Another important class of large-scale DML is based on primal-dual methods, i.e., ADMM [16], for which codes have seldom been studied. Thus, coding for ADMM based large-scale DML should be developed to combat straggler nodes, reduce communication loads and increase efficiency.

Despite the progress in coding for straggler nodes [4,8,9,10,11,12,13,14,24,25], the results are still preliminary and there are also various critical challenges for exploiting the advantages of DML, especially for *large-scale learning*: (1) Reliability and complexity—though coding has been proposed for addressing the straggler nodes to improve reliability, the existed schemes are mainly for the systems with a limited number of nodes or data. The coded DML schemes based on existing optimal error-control codes (i.e., maximum distance separable: MDS codes) [4,24,25] have very high encoding/decoding complexity when the number of involved nodes or the data volume scales up. Moreover, MDS codes treat every coding node equally and are not optimal for heterogeneous networks (e.g., IoT or mobile networks). (2) Communication loads—with increasing nodes or data volumes, the communication loads will quickly increase for exchanging model updates among learning nodes. Thus, coding schemes efficient in communication loads are critical for large-scale DML. (3) Limited learning functions—most of the existing coding schemes for DML are for gradient descend (primal method), i.e., combining coding with matrix multiplication and/or data shuffling [4,8,9,10,11,12,13,14,24,25]. Coding for many other important distributed learning functions, e.g., primal–dual optimization functions (also may be non-smooth or non-convex) in ADMM has seldom been explored. Moreover, existing coding for DML often runs in a master–worker structure, which may not be efficient (or even infeasible) for certain applications, e.g., those without master nodes. Thus, coding for fully decentralized DML should be also investigated. By encoding the messages to (or/and from) different destinations/sources in intermediate nodes, network coding shows the benefits of reducing information flow in the networks [29,30]. Moreover, it has been shown that network coding can improve the reliability and security of communication networks [12,31,32]. Thus, it is also valuable to discuss the applications of network coding to DML.

In what follows, we first introduce the basics on DML in Section 2. Then we discuss how error-control coding can help with the straggler problem in Section 3, the random coding construction in Section 4, and coding for primal–dual-based DML (ADMM) in Section 5. Finally, conclusions and discussion for future works are given in Section 6.

## 2. Introduction of Distributed Machine Learning

In general, DML will have two steps: (1) Agents learn local models from local data, maybe combining with global models. This step may iterate multiple rounds, i.e., local iterations, to produce a local model. (2) With local models, agents will reach consensus. These two steps may also iterate multiple rounds, i.e., global iterations. There are also different methods to implement the two steps, for instance, the primal and primal–dual methods as mentioned above. There are different ways to achieve consensus, for instance, through a central server, i.e., master–slave method or fully decentralized. For the former, the implementation is relatively straightforward. Yet, for the latter, there are also different approaches as will be discussed later on. For Step (1), the common local learning machine includes, for example, linear (polynomial) regressions, classification and neural networks. The common approach of these learning algorithms is to find the model parameters (e.g., weights in neural networks) that minimize the cost functions (such as mean-squared errors/L2 loss, hinge loss and cross-entropy loss). In general, convex cost functions should be chosen. For instance, for linear regression, we assume x,y as the input and output of the training data, respectively, and *w* (normally a matrix or a vector) as the weight to be optimized. If the mean-squared error cost functions are used, then the learning machine works as
(1)$$\min_{w}{\left\|xw-y\right\|}^{2}.$$

To find the optimal *w*, one common approach is to use gradient descend, which is a first-order iterative optimization algorithm for finding a local minimum of a differentiable function. If the cost function is convex, then the local minimum is also the global minimum [33]. For instance, in the training process of neural networks, gradient descend is commonly used to find the optimized weight and bias iteratively. The gradient is found by partial derivative of cost functions relative to optimizing variables (weight and bias of training examples). For instance, for node *i*, the optimizing variables can be updated by
(2)$$w_{t+1}^{i}=w_{t}^{i}-\gamma \nabla F\left(w_{t}^{i},D_{i}\right),$$
where *t* is the iteration step index, γ is the step size, $D_{i}$ is the data set (training samples) in node *i*, $F(w_{t}^{i})$ is the cost function with current optimizing variables, and $\nabla F(w_{t}^{i},D_{i})$ denotes the gradients for given $\left(w_{t}^{i},D_{i}\right)$ (by partial derivatives). The training process is normally performed in batches of data. $D_{i}$ can be further divided into subsets, e.g., *N* subsets, i.e., $D_{i}=\left\{D_{i}^{1},D_{i}^{2},\cdots ,D_{i}^{N}\right\}$. If subsets are exclusive, the gradients from different subsets are independent, i.e., $\nabla F\left(w_{t}^{i},D_{i}\right)=\left\{\nabla F\left(w_{t}^{i},D_{i}^{1}\right),\nabla F\left(w_{t}^{i},D_{i}^{2}\right),\cdots ,\nabla F\left(w_{t}^{i},D_{i}^{N}\right)\right\}$. However, in many DML systems, e.g., those based on MAPReduce file systems, or sensor nodes in neighboring areas, there may be overlapping data subsets, i.e., $D_{i}^{k}=D_{j}^{n}$ for certain k,n and i≠j. Therefore, there may be identical gradients in different nodes. These properties were recently exploited for coding. It it clear from (Equation 2) that for given gradients, the steps of finding optimal parameters are mainly linear matrix operations (matrix multiplications). Actually, in addition to neural networks, one core operation of many other learning algorithms is also matrix multiplications, such as regression, power-iteration-like algorithms, etc. [4]. Thus, one of the major coding schemes for DML is based on the matrix multiplication of the learning process [4,8,9,10,11,12,13,14,24,25]. Clearly, major coding schemes (forward error-control coding and network coding) are linear in terms of encoding and decoding operations, i.e., C=M×W, where *C*, *M* and *W* are codeword (vectors), coding matrix and information message, respectively. Since both learning and coding operations are linear matrix operations, then the coding matrix and learning matrix can be *jointly* optimized. On the other hand, coding can be optimized to provide efficient and reliable information pipelines for DML systems. In such way, coding and DML matrices are *separately* optimized. Separate optimization actually has been widely studied for many years for existing systems due to the simpler design relative to joint design. There are many works in the literature on the separate optimization of learning systems and coding schemes. We will focus on joint design in this article.

## 3. Coding for Reliable Large-Scale DML

In this section, we will first give a review on the basic principles of coding for reliable DML. Then, we will discuss two optimal construction of codes for DML.

One toy example of how coding can help to deal with stragglers can be found in Figure 1 [34]. For instance, it can be a federated learning network with worker and server nodes. There is partial overlapping for data segments in different worker nodes and thus the partial overlapping of gradients. As in Figure 1, we divide the data set of a node into multiple smaller sets to denote the partial overlapping of different nodes. Meanwhile, multiple sets in a node are also necessary for encoding as shown in the figure since one data set corresponds to one source symbol of the code. In the server node, a weight sum of the gradient is needed. In the figure, three worker nodes have different data parts of $D_{1},D_{2},D_{3}$, which are used to compute gradients $G_{1},G_{2},G_{3}$, respectively. In the server, an individual gradient is not needed but only their sum $G_{s}=G_{1}+G_{2}+G_{3}$. We can easily see that gradients from *any* two nodes can calculate $G_{s}$. For instance, if worker3 is outage, then $G_{s}=2\left(G_{1}/2+G_{2}\right)-\left(G_{2}-G_{3}\right)$ with two transmission coded blocks from worker1 and worker2. If there is no coding, then worker1 and worker2 have to transmit $G_{1},G_{2},G_{3}$ separately with three blocks after the coordination operations. Thus, coding can save the transmission and also coordination loads.

Though the idea of applying coding for DML is straightforward as shown in the above toy example, the code design will be rather challenging for large-scale DML, i.e., when the numbers of nodes and/or gradients per node are very large. One big challenge is how to construct encoding and decoding matrices, especially with limited complexity. In what follows, we will first give a brief introduction of the MAPReduce file systems, which are often used in DML. Then, we will discuss the coding schemes with deterministic construction [34]. The random construction based on fountain codes is given in the next section, which normally has lower complexity [13,14].

In large DML systems, MAPReduce is a commonly used distributed file storage system. As shown in Figure 2, there are three stages for the MAPReduce file systems: map, shuffling and reduce. In the system, data are stored in different nodes. In the map stage, stored data are sent to different computing nodes (e.g., cloud computing nodes), according to pre-defined protocols. In the shuffling stage, the computed results (e.g., gradients) are exchanged among nodes. Finally, the end users will collect the computed results in the reduce stage. MAPReduce can be used in federated learning, which was originally proposed for the applications in mobile devices [5]. In such a scenario, data are first sent to different worker nodes in the map stage, according to certain design principles. Then in the shuffling stage, local model parameters are aggregated in the server node. Finally, the aggregated models are obtained in the final iteration at the server. In such a way, worker nodes have all necessary data for computing local models, sent from storage nodes. However, there may be straggling worker nodes, due to either slow computing at the node or transmission errors in the channels. In such scenario, gradient coding [34] can be used to correct the straggler nodes.

To construct gradient coding, we use *A* to denote the possible straggler pattern multiplied by the corresponding decoding matrix, and *B* to denote how different gradients (or model parameters) are combined in the worker node. Thus, *A* denotes *transmission matrix multiplied by decoding* matrices in some sense (as they recover transmitting gradients from received coded symbols) and *B* can also be regarded as an *encoding* matrix. Assuming that *k* is the number of different gradients (data partitions) in all nodes and there are a total of *n* output channels in all nodes, the dimension of *B* is n×k. Denoting $\bar{g}={\left[g_{1},g_{2},\cdots ,g_{k}\right]}^{T}$ as the vector of all gradients, then worker node *i* transmits $b_{i}\bar{g}$, where $b_{i}$ is the *i*-th row of *B* and the encoding vector at node *i*. The dimension of *A* is k×n. A row of *A* corresponds to an instance of straggling patterns, in which 0 means a straggler node and how the gradients are reproduced in the server. Thus, all rows in *A* denote all possible ways of straggling. Denoting *f* as the number of surviving workers (none-stragglers), there are at most n−f 0s in each row of *A*. In the example of Figure 1, we only need the sum of gradients from worker nodes instead of the values of individual gradients. Thus, we have $AB=1_{k\times k}$ and each row of $AB\bar{g}$ is identically $G_{1}+G_{2}+G_{3}$, where $1_{k\times k}$ denotes all 1 matrix. For the example, we can easily see that
(3)$$A=\left\{\begin{array}{ccc} 0 & 1 & 2\\ 1 & 0 & 1\\ 2 & -1 & 0 \end{array}\right\},\;\mathrm{and}\;B=\left\{\begin{array}{ccc} 1/2 & 1 & 0\\ 0 & 1 & -1\\ 1/2 & 0 & 1 \end{array}\right\}.$$

Clearly, if we want individual values of $\bar{g}$, we should redesign A,B such that AB is an identity matrix. Or if we want the weighted sum of gradients (weights more general than 1), A,B should be also redesigned. From the description, we can see that the main challenge of designing the gradient coding is to find suitable encoding matrix *B* such that it can correct the straggling loss defined by *A*. In [34], two different ways of finding *B* and corresponding *A* are given, i.e., fractional repetition and cyclic repetition schemes as detailed in the following.

We denote *n* and *s* as the number of worker nodes and straggler nodes, respectively, and assume *n* is a multiple of s+1. Then, fractional repetition construction is described as the following steps.

Divide *n* workers into s+1 groups of size n/(s+1);In each group, divide all the data equally and disjointly, assigning s+1 partitions to each worker;All the groups are replicas of each other;After local computing, every worker transmits the sum of its partial gradient.

By the second step, in a group, the first worker obtains the first s+1 partitions from the map stage and computes the first s+1 gradients, and the second worker obtains the second s+1 partition from the map stage and computes the second s+1 gradient and so on. The encoding of each group of workers can be denoted by a block matrix ${\bar{B}}_{block}\left(n,s\right)\in R^{\frac{n}{s+1}\times n}$ with
(4)$${\bar{B}}_{block}\left(n,s\right)={\left[\begin{array}{cccc} 1_{1\times (s+1)} & 0_{1\times (s+1)} & \cdots & 0_{1\times (s+1)}\\ 0_{1\times (s+1)} & 1_{1\times (s+1)} & \cdots & 0_{1\times (s+1)}\\ \vdots & \vdots & \ddots & \vdots \\ 0_{1\times (s+1)} & 0_{1\times (s+1)} & \cdots & 1_{1\times (s+1)} \end{array}\right]}_{\frac{n}{s+1}\times n}.$$

Here $1_{1\times (s+1)}$ and $0_{1\times (s+1)}$ means 1×(s+1) matrix of all 1 s and all 0 s (row vector), respectively. Then B is obtained by replicating s+1 copies of ${\bar{B}}_{block}\left(n,s\right)$, i.e.,
(5)$$B=B_{frac}=\left[\begin{array}{c} {\bar{B}}_{block}^{1}\left(n,s\right)\\ {\bar{B}}_{block}^{2}\left(n,s\right)\\ \vdots \\ {\bar{B}}_{block}^{\left(s+1\right)}\left(n,s\right) \end{array}\right],$$
where ${\bar{B}}_{block}^{i}\left(n,s\right)={\bar{B}}_{block}\left(n,s\right),$ for i∈{1,⋯,s+1}. In addition to the encoding matrix $B_{frac}$, reference [34] also gives the algorithms of constructing the corresponding *A* matrix as follows.

It was shown in [34] that by fractional repetition schemes, $B=B_{frac}$ from (Equation 5) and *A* from Algorithm 1 can correct any *s* straggler. It can be more formally stated as the following theorem.
**Algorithm 1** Algorithm to compute *A* for fractional repetition coding.**Input:***B* = *B_frac_*;*f* ← binom(*n*,*s*)  *A* ← zeros(*f*,*n*)  **for**
*I* ⊆ [*n*], *s.t.*∣*I*∣ = (*n* − *s*) **do**⌊ *a* = zeros(1,*k*)  *x* = ones(1,*k*)/*B*(*I*,:)  *a*(*I*) = *x*  *A* = [*A*;*a*]**output:***A* s.t. *AB* = 1_f×k_;

**Theorem** **1.**
*Consider $B=B_{frac}$ from (Equation 5) for a given number of workers n and stragglers s(<n). Then, the scheme $\left(A,B_{frac}\right)$, with A from Algorithm 1 is robust to any s straggler.*


Here, we refer the interested readers to [34] for the proof. In addition to fractional repetition construction, another way of finding the *B* matrix is the cyclic repetition scheme, which does not require *n* to be a multiple of s+1. The algorithm to construct the cyclic repetition *B* matrix is given as follows.

Actually, the resultant matrix $B=B_{cyc}$ from Algorithm 2 has the following support (non-zero parts):(6)$$supp\left(B_{cyc}\right)=\left[\begin{array}{cccccccccc} {}* & * & \cdots & * & * & 0 & 0 & \cdots & 0 & 0\\ 0 & * & * & \cdots & * & * & 0 & \cdots & 0 & 0\\ \vdots & \vdots & \vdots & \vdots & \vdots & \vdots & \ddots & \ddots & \vdots & \vdots \\ 0 & 0 & \cdots & 0 & 0 & * & * & \cdots & * & *\\ \vdots & \vdots & \vdots & \vdots & \vdots & \vdots & \ddots & \ddots & \vdots & \vdots \\ & \cdots & * & * & 0 & 0 & \cdots & 0 & 0 & * \end{array}\right],$$
where ∗ is the non-zero entries in $B_{cyc}$, and in each row of $supp(B_{cyc})$, there are (s+1) non-zero entries. The position of non-zero entries is right shifted one step and cycled around until the last row. The construction of *A* matrix follows Algorithm 1 also for $B_{cyc}$. It was shown in [34] that cyclic repetition schemes can also correct any *s* stragglers:
**Algorithm 2** Algorithm to construct $B=B_{cyc}$.**Input:***n*,*s*(<*n*)*H* = binom(*n*,*s*)  *H* = −sum(*H*(:,1:*n*−1),2)  *B* = zeros(*n*)  **for**
*i* = 1:*n*
**do**
⌊ *j* = mod(*i* − 1:*s* + *i* − 1, *n*) + 1  *B*(*i*,*j*) = [1; −*H*(:,*j*(2 : *s* + 1))]\*H*(:,*j*(1))]**output:**$B\in R^{n\times n}$ with (*s* + 1) non-zeros in each row.

**Theorem** **2.**
*Consider $B=B_{cyc}$ from Algorithm 2, for a given number of workers n and stragglers s(<n). Then, the scheme $\left(A,B_{cyc}\right)$, with A from Algorithm 1 is robust to any s straggler.*


Fractional repetition and cyclic repetition schemes provide specific methods of encoding and decoding for master–worker DML for tolerating any *s* stragglers. More generally, it was also shown in [34] the necessary conditions for matrix *B* for tolerating any *s* stragglers if the following conditions are satisfied.

Condition 1 (B-Span): Consider any scheme (A,B) robust to any *s* stragglers, given n(s<n) workers, then every subset $\left(I\right)\subseteq \mathrm{span}\{b_{i}|i\in \left(I\right)\}$ is satisfied, where span {·} is the span of vectors.

If *A* matrix is constructed by Algorithm 1, (A,B) with Condition 1 is also sufficient.

**Corollary** **1.**
*If A matrix is constructed by Algorithm 1 and B satisfies Condition 1, (A,B) can correct any s stragglers.*


Numerical results: In Figure 3, the average time per iteration for different schemes is compared from [34]. In *naive scheme*, the data are divided uniformly across all workers without replication, and the master just waits for all workers to send their gradients. In *ignoring the s straggler scheme*, the data distribution is the same as the naive scheme. However, the master node only waits until n−s worker nodes successfully send their gradients (no need to wait for all gradients). Thus, as discussed in [34], ignoring the straggler scheme may lose in the generalization performance by ignoring a part of data sets of straggler nodes. The running learning algorithms are based on logistic regression. The training data are from the Amazon Employee Access dataset from Kaggle. The delay is introduced by the computing latency of AWS clusters, and there is no transmission error. As shown in the figure, the naive scheme performs the worst. With increasing stragglers, coding schemes also perform better than ignoring straggler schemes as expected.

## 4. Random Coding Construction for Large-Scale DML

The gradient coding in [34] works well for the DML scheme with a master–worker structure with limited sizes (finite number of nodes and limited data partitions). However, the deterministic construction of encoding and decoding matrices may be challenging when the number of nodes or data partitions (e.g., *n* or *k*) is large. The first challenge is the complexity of encoding and decoding, both of which are based on matrix multiplication, which may be rather complex, especially for decoding (e.g., based on Gaussian elimination). Though DML with MDS codes is optimal in terms of code distance (i.e., the degree of tolerance to the amount of straggler nodes), the coding complexity will be rather high with the increasing number of participating nodes, i.e., for hundreds or even thousands of computing nodes. For instance, Reed–Solomon codes normally need to run in non-binary fields, which are of high complexity. Another challenge is lack of flexibility. Both factional repetition and cyclic repetition coding schemes assume static networks (worker nodes and data). However, in practice, the participating nodes may be varying in mobile nodes or sensors, for example. In the mobile computing scenario, the number of participating nodes may be unknown. It will rather difficult to design deterministic coding matrices (*A* or *B*) in such a scenario. Similarly, if the data are from sensors, the amount of data may also be varying. Thus, the deterministic construction of coding is hard to adapt to these scenarios, which, however, are very common in large-scale learning networks. Thus, coding schemes efficient in varying networks and of low complexity are preferable for large-scale DML. In [13,14], we investigated the random coding for DML (or distributed computing in general) to address the problems. Our coding scheme is based on fountain codes [35,36,37]. The coding scheme is introduced as follows.

*Encoding Phase:* As shown in Figure 4, we consider a network with multiple storage and computing/fog nodes. Let $FN_{f}$ denote the *f*-th fog node and let $SU_{s}$ denote the *s*-th storage unit with f∈{1,2,⋯,F} and s∈{1,2,⋯,S}, respectively. Let $D_{f}$ denote the dataset node *f* needed to finish a learning task. $D_{f}$ will be obtained from the storage units available to node *f*. For instance, in a DML with wireless links as in Figure 4, $D_{f}$ means the data union for all the storage units within the communication range of $FN_{f}$ (i.e., within $R_{f}$). Similar to federated learning, $FN_{f}$ will use the current model parameters to calculate gradients, namely, intermediate gradients, denoted as $g_{f}=\left[g_{f,1},g_{f,2},\cdots ,g_{f,|D_{f}|}\right]$, where $g_{f,a}$ means the gradient trained by data $a(a\in D_{f})$ and $|D_{f}|$ is the size of $D_{f}$. Meanwhile, fog nodes need to calculate the intermediate model parameters (e.g., weight) $w_{f}=\left[w_{f,1},w_{f,2},\cdots ,w_{f,|w_{f}|}\right]$, where $|w_{f}|$ is the length of model parameters learned at $FN_{f}$. Then the intermediate gradients and model parameters will be sent out to other fog nodes (or the central sever if there is one) for further processing after encoding. The coding process for $g_{f}$ is as follows.

A number $d_{g}$ is selected according to degree distribution $\Omega \left(x\right)=\sum_{d_{g}=1}^{|D_{f}|}\Omega_{d_{g}}x^{d_{g}}$ with probability $\Omega_{d_{g}}x^{d_{g}}$;Then, $d_{g}$ intermediate gradients are selected uniformly at random from $g_{f}$ to encode into one coded intermediate gradient;The above two steps repeated until $Q_{f}^{g}=\left(1+\eta_{f}\right)\left|D_{f}\right|$ coded intermediate gradients are formed, where $\eta_{f}\left(\geq 0\right)$ is the expanding coefficient of the fountain codes (denoting redundancy).

Ω(x) can be optimized by the probability of straggling (regarded as erasure) due to channel errors, slow computing, etc. The optimization of the degree distribution for distributed fountain codes can be found in, for example, [38], and we will not discuss it here for space limitation. With the above coding process, the resulted coded intermediate gradients are
(7)$$c_{f}^{g}=\left[g_{f,1},g_{f,2},\cdots ,g_{f,|D_{f}|}\right]G_{f}^{g}=g_{f}G_{f}^{g},$$
where $G_{f}^{g}$ is the generator matrix at fog node $FN_{f}$. The encoding process for $w_{f}$ is the same as that of $g_{f}$ with a possibly different degree distribution $\mu \left(x\right)=\sum_{d_{w}=1}^{w_{f}}\mu_{d_{w}}x^{d_{w}}$. The formed $Q_{f}^{w}=\left(1+\eta_{f}\right)w_{f}$ coded intermediate parameters can be written as $c_{f}^{w}=w_{f}G_{f}^{w}$, where $G_{f}^{w}$ is the generator matrix at $FN_{f}$ for model parameters.

*Exchanging Phase:* The coded intermediate gradients $c_{f}^{g}$ and model parameters $c_{f}^{w},\left(f\in \left\{1,2,\cdots ,N\right\}\right)$ are exchanged among fog nodes. Let *M* be the total number of all different data in all *F* nodes, $M\leq \sum_{f=1}^{F}\left|D_{f}\right|$. The equality holds only if *F* datasets are disjoint.

*Decoding Phase:* The generator matrices for the received coded intermediate gradients and model parameters from fog node $FN_{i}\left(i\in \left\{1,2,\cdots ,F\right\}\right)\backslash \left\{f\right\}$ at $FN_{f}$ are ${\tilde{G}}_{i,f}^{g}$ with size $\left|G\right|\times Q_{i,f}^{g}$ and ${\tilde{G}}_{i,f}^{w}$ with size $w_{i}\times Q_{i,f}^{w}$, respectively, where $Q_{i,f}^{g}=\left(1-\epsilon_{i,f}\right)Q_{i}^{g}$ and $Q_{i,f}^{w}=\left(1-\epsilon_{i,f}\right)Q_{i}^{w}$. Here $\epsilon_{i,f}$ denotes the straggling probability from $FN_{i}$ to $FN_{f}$ due to various reasons, e.g., physical-layer erasure, slow computing, and congestion. Thus, the generator matrices corresponding to the received coded intermediate gradient and model parameters at $FN_{f}$ can be written as ${\tilde{G}}_{f}^{g}=[1_{1}{\tilde{G}}_{1,f}^{g},\cdots ,1_{f-1}{\tilde{G}}_{f-1,f}^{g},1_{f+1}{\tilde{G}}_{f+1,f}^{g},\cdots ,1_{F}{\tilde{G}}_{F,f}^{g}$ and ${\tilde{G}}_{f}^{g}=[1_{1}{\tilde{G}}_{1,f}^{g},\cdots ,1_{f-1}{\tilde{G}}_{f-1,f}^{g},1_{f+1}{\tilde{G}}_{f+1,f}^{g},\cdots ,1_{F}{\tilde{G}}_{F,f}^{g}$ and ${\tilde{G}}_{f}^{w}=[1_{1}{\tilde{G}}_{1,f}^{w},\cdots ,1_{f-1}{\tilde{G}}_{f-1,f}^{w},1_{f+1}{\tilde{G}}_{f+1,f}^{w},\cdots ,1_{F}{\tilde{G}}_{F,f}^{w}$, respectively. Here $I=\{1_{1},\cdots ,1_{F}\}$ is an indicator parameter. Let λ be the probability of straggling. Then, $I_{f},\left(f\in \left\{1,2,\cdots ,F\right\}\right)$ can be evaluated as
(8)$$I_{f}=\left\{\begin{array}{ccc} 1, & \mathrm{with}\;\mathrm{probability} & 1-\lambda ,\\ 0, & \mathrm{with}\;\mathrm{probability} & \lambda . \end{array}\right.$$

Then fog node $FN_{f}$ decodes the received coded intermediate parameters from ${\tilde{G}}_{i,f}^{g}$ and ${\tilde{G}}_{i,f}^{w},\left(i\in \left\{1,2,\cdots ,F\right\}\backslash \left\{f\right\}\right)$, and tried to decode $N-|D_{f}|$ new gradients and $\Gamma_{w}\sum_{i\in \{1,2,\cdots ,F\}\backslash \{f\}}w_{i}$ model parameters, where $\Gamma_{w}\in \left[0,1\right]$ is a parameter determined by specific learning algorithms. For the benefits of fountain codes (e.g., LT or Raptor codes), the iterative decoding is feasible if the numbers of received coded gradients or model parameters are slightly larger than those of gradients and models in transmitting fog nodes. Clearly, to optimize the code degree distribution and task allocation, it is critical for a node to know the number of received intermediate gradients and model parameters at the node. For the purpose, we have the following analysis.

Assume $\gamma_{a,b}$ as the overlapping ratio of the dataset in $FN_{a}$ and $FN_{b}$, then for all fog nodes, we have the overlapping ratio as follows:(9)$$\gamma =\left[\begin{array}{cccc} 1 & \gamma_{1,2} & \cdots & \gamma_{1,F}\\ \gamma_{2,1} & 1 & \cdots & \gamma_{2,F}\\ \vdots & \vdots & \ddots & \vdots \\ \gamma_{F,1} & \gamma_{F,2} & \cdots & 1 \end{array}\right].$$

If $\gamma_{a,b}=0$, then node $FN_{a}$ and $FN_{b}$ has disjoint datasets. At $FN_{f}$, $|D_{f}|$ intermediate gradients are known. Thus, $A=N-|D_{f}|$ new intermediate gradients are required for updating model parameters $w_{f}$. Then, we have the following result:

**Theorem** **3.**
*The total number of new intermediate gradients received from the other fog nodes at $FN_{f}$ can be calculated by $\Delta =\sum_{\pi_{i},i=1}^{F-1}1_{\pi_{i}}\left(\left(1-\gamma_{\pi_{i},f}\right)\varphi \left(i,f\right)\right)\cdot \left|D_{\pi_{i}}\right|$, where φ(i,f) can be written as*

(10)
$$\varphi \left(i,f\right))=\left\{\begin{array}{ccc} 1, & if & i=1,\\ \Pi_{a=1}^{i-1}\left(1-\gamma_{\pi_{i},\pi_{i}-\pi_{a}}|\Theta_{a,f}\right), & if & 2\leq i\leq F-1, \end{array}\right.$$

*where $\Theta_{a,f}$ is a set formed by the indices of fog nodes, and it can be evaluated by*

(11)
$$\Theta_{a,f}=\left\{\begin{array}{ccc} \{f\}, & \mathrm{if} & a=1,\\ \{f,\pi_{1},\cdots ,\pi_{a-1}\}, & \mathrm{if} & a>1. \end{array}\right.$$



If γ is known at each fog node (or at least from the transmitted neighbors at each receiving node), then Δ can be evaluated, and the computation and communication loads can be optimized through proper task assignment and code degree optimization. Theorem 3 is for gradients, and a similar analysis also holds for model parameters. In Figure 5, we show the coding gains in terms of communication loads, which are defined as the ratio of the total number of data transmitted by all the fog nodes to the data required at these fog nodes. As we can see from the figure, if the number of nodes *F* or straggler probability increases, the coding gains increase as expected.

We note that both deterministic codes in Section 3 and random construction coding here are actually a type of network coding [29,30], which can reduce communication loads by computing at intermediate nodes (fog nodes) [3,4]. More recently, one type of special network codes, i.e., BATS (batched sparse) codes, was proposed with two layered codes as shown in Figure 6. For outer codes, we can use error control codes such as fountain codes in MAP phase. For inner codes, network codes can be used such as random linear network codes in data shuffling stage. In [12], we studied BATS codes for fog computing networks. As shown in Figure 7, numerical results demonstrate that the BATS codes can achieve a lower communication load than uncoded and deterministic codes (network codes) if the computing load is lower than certain thresholds. Here, we skip further details and refer interested readers to [12].

## 5. Coding for ADMM

### 5.1. Introduction and System Setup

As a primal–dual optimization method, ADMM is shown to be able to generally converge at a rate of O(1/t) for convex functions, where *t* is the iteration number [16], which is often faster than the schemes based on primal methods. Meanwhile, ADMM also has the benefits of robustness to non-smooth/non-convex functions and being adaptive to fully decentralized implementation. Thus, ADMM is especially suitable for large-scale DML and has attracted substantial research interests. For DML, especially for the fully decentralized learning system without a central server, we can denote the learning network as G=(N,E), where N={1,…,N} is the set of agents (computing nodes) and E is the set of links. For ADMM, agents aim at solving the following consensus optimization problem collaboratively:(12)$$\min_{x}\;\sum_{i=1}^{N}f_{i}\left(x;D_{i}\right),$$
where $f_{i}:R^{p}\to R$ is the local optimization function of agent *i*, and $D_{i}$ is the data set of agent *i*. All the agents share a global optimization variable $x\in R^{n}$. Data sets of different agent may have overlapping, i.e., $D_{i}\cap D_{j}\neq \emptyset$, for a part or all i≠j. This can happen, for instance, among the sensors of nearby areas for weathers, traffic, smart grids, etc., or if MAPReduce is used, the same data are mapped to different agents. For ADMM, (Equation 12) is solved iteratively by a two-step process:Step (a), local optimization of $f_{i}$ on receiving updated global variable and with $D_{i}$ (normally by augmented Lagrangian as detailed below);Step (b), global variable *x* reaches consensus.

With DML, there are also straggler nodes and unreliable-link challenges for ADMM, especially for large-scale and heterogeneous networks or with wireless links. However, with primal–dual optimization, it is very hard (if possible) to transfer ADMM optimization process into a linear function (e.g., matrix multiplication as in gradient descend). Thus, coding schemes based on linear operations (e.g., matrix multiplication in [4,8,9,10,11,24,25]) are impossible to be directly used in ADMM and there are very few results on coding for ADMM so far, to our best knowledge. To address the problem, one solution is to use coding separately for two steps of ADMM. For instance, error control coding can be used for local optimization if the data are stored in different locations for an agent. For the global consensus, network coding can be used to reduce the communication loads and increase reliability. In [15], we preliminarily investigated how coding (MDS codes) can be used in local optimization (step (a)). A more detailed introduction is given as follows.

As depicted in Figure 8, a distributed computing system consists of multiple agents, each of which is connected with several edge computing nodes (ECNs). Agents can communicate with each other through links. ECNs are capable of processing data collected from sensors, and transferring desired messages (e.g., model updates) back to the connected agent. Based on the agent coverage and computing resources, the ECNs connected to agent i(∈N) are denoted as $K_{i}=\left\{1,\ldots ,K_{i}\right\}$. This model is common in current intelligent systems, such as smart factories or homes.

The multi-agent system seeks to find out the optimal solution $x^{*}$ by solving (Equation 12). $D_{i}$ is allocated to dispersed ECNs $K_{i}$. The formulation of decentralized optimization problem can be described as follows. By defining $x=\left[x_{1},\ldots ,x_{N}\right]\in R^{pN\times d}$ and introducing a global variable $z\in R^{p\times d}$, problem (Equation 12) can be reformulated as
(13)$$\begin{array}{c} \left(P-1\right):\min_{x,z}\;\sum_{i=1}^{N}f_{i}\left(x_{i};D_{i}\right),\;\;\;s.t.\;1⊗z-x=0, \end{array}$$
where $1={\left[1,\ldots ,1\right]}^{T}\in R^{N}$, and ⊗ is the Kronecker product. In the following, $f_{i}\left(x_{i},D_{i}\right)$ is denoted as $f_{i}\left(x_{i}\right)$ for simplifying illustration.

In what follows, we will present communication-efficient and straggler-tolerant decentralized algorithms, by which the agents can collaboratively find an optimal solution through local computations and limited information exchange among neighbors. In the scheme, local gradients are calculated in dispersed ECNs, while variables, including primal and dual variables and global variables *z*, are updated in the corresponding agent. For illustration purpose, we will first present stochastic ADMM (sI-ADMM) and then coded version of sI-ADMM (i.e., csI-ADMM). Both of them are proposed in [15]. The standard incremental ADMM iterations for decentralized consensus optimization will be reviewed first. The augmented Lagrangian function of problem (P-1) is
(14)$$L_{\rho }\left(x,y,z\right)=\sum_{i=1}^{N}f_{i}\left(x_{i}\right)+\langle y,1⊗z-x\rangle +\frac{\rho }{2}{∥1⊗z-x∥}^{2},$$
where $y=\left[y_{1},\ldots ,y_{N}\right]\in R^{pN\times d}$ is the dual variable, and ρ>0 is a penalty parameter. With incremental ADMM (I-ADMM) [39,40], with guaranteeing $\sum_{i=1}^{N}\left(x_{i}^{1}-\frac{y_{i}^{1}}{\rho }\right)=0$ (e.g., initialize $x_{i}^{1}=y_{i}^{1}=0$), the updates of x, y and *z* at the (k+1)-th iteration follow:
(15a)$$x_{i}^{k+1}:=\left\{\begin{array}{c} \arg\min_{x_{i}}\;f_{i}\left(x_{i}\right)+\frac{\rho }{2}{∥z^{k}-x_{i}+\frac{y_{i}^{k}}{\rho }∥}^{2},\;i=i_{k};\\ x_{i}^{k},\;\mathrm{otherwise}; \end{array}\right.$$
(15b)$$y_{i}^{k+1}:=\left\{\begin{array}{c} y_{i}^{k}+\rho \left(z^{k}-x_{i}^{k+1}\right),\;i=i_{k};\\ y_{i}^{k},\;\mathrm{otherwise}; \end{array}\right.$$
(15c)$$\begin{array}{cc} \; & z^{k+1}:=z^{k}+\frac{1}{N}\left[\left(x_{i_{k}}^{k+1}-x_{i_{k}}^{k}\right)-\frac{1}{\rho }\left(y_{i_{k}}^{k+1}-y_{i_{k}}^{k}\right)\right]. \end{array}$$

For ADMM, solving augmented Lagrangian especially for the *x*-update above may lead to rather high computational complexity. To achieve fast computation for *x*-update, *first-order* approximation and *mini-batch stochastic* optimization in (15a) can be adapted. Furthermore, a quadratic proximal term with parameter $\tau^{k}$ is proposed in [15] to stabilize the convergence behavior of the inexact augmented Lagrangian method. Ref. [15] also introduces the updating step-size $\gamma^{k}$ for the dual update. Both parameters $\tau^{k}$ and $\gamma^{k}$ can be adjusted with iteration *k*. Finally, the updates of x and y at the (k+1)-th iteration are presented as follows:
(16a)$$x_{i}^{k+1}:=\left\{\begin{array}{c} \arg\min_{x_{i}}\;\langle G_{i}\left(x_{i}^{k};\xi_{i}^{k}\right),x_{i}-x_{i}^{k}\rangle +\langle y_{i}^{k},z^{k}-x_{i}\rangle \\ \;+\frac{\rho }{2}{∥z^{k}-x_{i}∥}^{2}+\frac{\tau^{k}}{2}{∥x_{i}-x_{i}^{k}∥}^{2},\;i=i_{k};\\ x_{i}^{k},\;\mathrm{otherwise}; \end{array}\right.$$
(16b)$$y_{i}^{k+1}:=\left\{\begin{array}{c} y_{i}^{k}+\rho \gamma^{k}\left(z^{k}-x_{i}^{k+1}\right),\;i=i_{k};\\ y_{i}^{k},\;\mathrm{otherwise}; \end{array}\right.$$
where $G_{i}\left(x_{i}^{k};\xi_{i}^{k}\right)$ is the mini-batch stochastic gradient, which can be obtained through $G_{i}\left(x_{i}^{k};\xi_{i}^{k}\right)=\frac{1}{M}\sum_{l=1}^{M}\nabla F_{i}\left(x_{i}^{k};\xi_{i,l}^{k}\right)$. To be more specific, *M* is the mini-batch size of sampling data, $\xi_{i}^{k}={\left\{\xi_{i,l}^{k}\right\}}_{M}$ denotes a set of independent and identically distributed randomly selected samples in one batch, and $\nabla F_{i}\left(x_{i}^{k};\xi_{i,l}^{k}\right)$ corresponds to the stochastic gradient of a single example $\xi_{i,l}^{k}$.

### 5.2. Mini-Batch Stochastic I-ADMM

For above setup of ADMM, *response time* is defined as the execution time for updating all variables in each iteration. In the updates, all steps, including *x*-update, *y*-update and *z*-update, are assumed to be in agents rather than ECNs. In practice, the update is often computed in a tandem order, which leads to a long response time. With the fast development of edge/fog computing, it is feasible to further reduce the response time since computing the local gradients can be dispersed to multiple edge nodes, as shown in Figure 8. Each ECN computes a gradient using local data and shares the result with its corresponding agent, and no information is directly exchanged among ECNs. Agents can be activated in a predetermined circulant pattern, e.g., according to a Hamiltonian cycle, and ECNs are activated whenever the connected agent is active, as shown in Figure 8. A Hamiltonian cycle based activation pattern is a cyclic pattern through a graph that visits each agent exactly once (i.e., 1→2→4→5→3 in Figure 8). Correspondingly, the mini-batch stochastic incremental ADMM (sI-ADMM) [15] is presented in Algorithm 3. At agent $i_{k}$, global variable $z^{k+1}$ gets updated and is passed as a token to the next agent $i_{k+1}$ via a pre-determined traversing pattern, as shown in Figure 8. Specifically, in the *k*-th iteration with cycle index m=⌊k/N⌋, agent $i_{k}$ is activated. Token $z^{k}$ is first received and then the active agent broadcasts the local variable $x_{i}^{k}$ to its attached ECNs $K_{i}$. According to batch data with index $I_{i,j}^{k}$, new gradient $g_{i,j}$ is calculated in each ECN, followed by the gradient update, *x*-update, *y*-update and *z*-update in agent $i_{k}$, via steps 21–24 in Algorithm 3. At last, the global variable $z^{k+1}$ is passed as a token to its neighbor $i_{k+1}$. In Algorithm 3, the stopping criterion is reached when $∥z^{k}-x_{i}^{k}∥\leq \epsilon^{pri}\;\mathrm{and}\;∥G_{i}\left(x_{i}^{k};\xi_{i}^{k}\right)-y_{i}^{k}∥\leq \epsilon^{dual},\forall i\in N$, where $\epsilon^{pri}$ and $\epsilon^{dual}$ are two pre-defined feasibility tolerances.
**Algorithm 3** Mini-batch stochastic I-ADMM (sI-ADMM)1:**initialize**: $\left\{z^{1}=x_{i}^{1}=y_{i}^{1}=0,|i\in N\right\}$, batch size *M*;2:**Local Data Allocation:**3:**for** agent i∈N **do**4:    **divide** $D_{i}$ labeled data into $K_{i}$ equally disjoint partitions and denote each partition as $\xi_{i,j},j\in K_{i}$;5:    **for** ECN $j\in K_{i}$ **do**6:        **allocate** $\xi_{i,j}$ to ECN *j*;7:        **partition** $\xi_{i,j}$ examples into multiple batches with each size $M/K_{i}$;8:    **end for**9:**end for**10:**Updating Process:**11:**for**k=1,2,…**do**12:    **Steps of Active Agent $i=i_{k}=\left(k-1\right)\;\mathrm{mod}\;N+1$:**13:    **receive** token $z^{k}$;14:    **broadcast** local variable $x_{i}^{k}$ to ECNs $K_{i}$;15:    **ECN $j\in K_{i}$ computes gradient in parallel**:16:      **receive** local primal variable $x_{i}^{k}$;17:      **select** batch $I_{i,j}^{k}=m\;\mathrm{mod}\;\left\lfloor |\xi_{i,j}|\cdot K_{i}/M\right\rfloor$;18:      **update** gradient $g_{i,j}=\frac{K_{i}}{M}\sum_{l=1}^{\frac{M}{K_{i}}}\nabla F_{i}\left(x_{i}^{k};\xi_{i,l}^{k}\right)$;19:      **transmit** $g_{i,j}$ to the connected agent;20:    **until** the $K_{i}$-th responded message is received;21:    **update** gradient via gradient summation:
(17)$$G_{i}\left(x_{i}^{k};\xi_{i}^{k}\right)=\frac{1}{K_{i}}\sum_{j=1}^{K_{i}}g_{i,j};$$22:    **update** $x^{k+1}$ according to (16a);23:    **update** $y^{k+1}$ according to (16b);24:    **update** $z^{k+1}$ according to (15c);25:    **send** token $z^{k+1}$ to agent $i_{k+1}$ via link ($i_{k},i_{k+1}$);26:    **until** the stopping criterion is satisfied.27:**end for**

### 5.3. Coding for Local Optimization for sI-ADMM

With less reliable and limited computing capability of ECNs, straggling nodes may be a significant performance bottleneck in the learning networks. To address this problem, error control codes can be used to mitigate the impact of the straggling nodes by leveraging data redundancy. Similar to Section 3, two MDS-based coding methods over real field R, i.e., *fractional* repetition scheme and *cyclic* repetition scheme, can be adopted and integrated with sI-ADMM for reducing the responding time in the presence of straggling nodes. The coded sI-ADMM (csI-ADMM) approach is presented in Algorithm 4. Denote the minimum required ECNs number by $R_{i}$ and the maximum number of stragglers the system can tolerate by $S_{i}$. Different from sI-ADMM, in csI-ADMM, encoding and decoding processes are used in each ECN $j\in K_{i}$ and its corresponding agent *i*, respectively. $G_{i}\left(x_{i}^{k};\xi_{i}^{k}\right)$ will be updated via steps 15–20, where the local gradient is calculated in ECN $j\in K_{i}$ in parallel via selected $\left(S_{i}+1\right)\bar{M}/K_{i}$ batch samples, and the gradient summation can be recovered in active agent $i_{k}$ with the responded messages from any $R_{i}$ out of $K_{i}$ ECNs to combat slow links and straggler nodes. As in steps 22–26 of sI-ADMM, activated agent $i_{k}$ then updates local variables successively. Computation redundancy is introduced, but agent *i* can tolerate any $\left(S_{i}=K_{i}-R_{i}\right)$ stragglers.
**Algorithm 4** Coded sI-ADMM (csI-ADMM)1:**initialize**: $\left\{z^{1}=x_{i}^{1}=y_{i}^{1}=0|i\in N\right\}$, batch size $\bar{M}$;2:**Local Data Allocation:**3:**for**agenti∈N**do**4:    **divide** $D_{i}$ labeled data based on repetition schemes in [34] and denote each partition as $\xi_{i,j},j\in K_{i}$;5:    **for** $\mathrm{ECN}\;j\in K_{i}$ **do**6:        **allocate** $\xi_{i,j}$ to ECN *j*;7:        **partition** $\xi_{i,j}$ examples into multiple batches with each size $\left(S_{i}+1\right)\bar{M}/K_{i}$;8:    **end for**9:**end for**10:**Updating Process:**11:**for**k=1,2,…**do**12:    **Steps of Active Agent $i=i_{k}=\left(k-1\right)\;\mathrm{mod}\;N+1$:**13:    **run** steps 13–14 of Algorithm 314:    **ECN $j\in K_{i}$ computes gradient in parallel**:15:      **run** step 16 of Algorithm 316:      **select** batch
(18)$$I_{i,j}^{k}=m\;\mathrm{mod}\;\left\lfloor |\xi_{i,j}|\cdot K_{i}/\left(S_{i}+1\right)\bar{M}\right\rfloor ;$$17:      **update** $g_{i,j}$ via encoding function $p_{enc}^{j}\left(\cdot \right)$;18:      **transmit** $g_{i,j}$ to the connected agent;19:    **until** the $R_{i}$-th fast responded message is received;20:    **update** gradient via decoding function $q_{dec}^{i}\left(\cdot \right)$;21:    **run** steps 22–26 of Algorithm 3;22:**end for**

### 5.4. Simulations for Coded Local Optimization

Both computed-generated and real-world datasets are used to evaluate the performance of the coded stochastic ADMM algorithms. The experimental network G consists of *N* agents and $E=\frac{N(N-1)}{2}\eta$ links, where η is the network connectivity ratio. For agent *i*, $K_{i}=K$ ECNs with the same computing power (e.g., computing and memory) are attached. To reduce the impact of token traversing patterns, both the Hamiltonian cycle-based and non-Hamiltonian cycle-based (i.e., the shortest path cycle-based [41]) token traversing methods are evaluated for the proposed algorithms.

To demonstrate the advantages of the coding schemes, csI-ADMM algorithms are compared with uncoded sI-ADMM algorithms with respect to the accuracy [42], which is defined as
(19)$$\mathrm{accuracy}=\frac{1}{N}\sum_{i=1}^{N}\frac{∥x_{i}^{k}-x^{*}∥}{∥x_{i}^{1}-x^{*}∥},$$
where $x^{*}\in R^{p\times d}$ is the optimal solution of (P-1), and the test error [43], which is defined as the mean square error loss. For demonstrating the robustness against straggler nodes, distributed coding schemes, including *cyclic* and *fractional* repetition methods and the uncode method, are used for comparison. For fair comparison, the parameters for algorithms are tuned and kept the same in different experiments. Moreover, unicast is considered among agents, and the communication cost per link is 1 unit. The consumed time for each communication among agents is assumed to follow a uniform distribution $U(10^{-5},10^{-4})$ seconds. The response time of each ECN is measured by the computation time, and the overall response time of each iteration is equal to the execution time for updating all variables in each iteration. All experiments were performed using Python on an Intel CPU @2.3 GHz (16 GB RAM) laptop.

To show the benefit of coding, in Figure 9, we compare the accuracy vs. running time for both coded and uncoded sI-ADMM. In simulation, the maximum delay $\epsilon_{i},\left(i=1,2,3\right)$ for stragglers in each iteration is considered. For illustration purpose, we set up different $\epsilon_{i}$ with $\epsilon_{1}>\epsilon_{2}>\epsilon_{3}$ in simulation. For showing the benefits of coding to the convergence rate, convergence vs. straggler nodes trade-off for csI-ADMM, the impact of the number of straggler nodes on the convergence speed is shown in Figure 10. In simulations, 10 independent experiment runs are performed with the same simulation setup on synthetic data and take an average for presentation. We can see that, with an increasing number of straggler nodes, the convergence speed decreases. This is because increasing the number of straggler nodes decreases the allowable mini-batch size allocated in each iteration and therefore affects the convergence speed.

### 5.5. Discussion

Above, we discuss the application of error-control coding in the local optimization step of ADMM. In the agent consensus step, there are also straggling or transmission errors for updating global variables. To improve reliability in the consensus step, we can use linear network error correction codes [31] or BATS codes [32] based on LT codes. For the latter, the global variable (vector) is divided into many smaller vectors. The encoding process continues until certain stopping criteria are reached (e.g., feedback from other nodes or time out). There are quite a few papers on applying network coding for consensus; see [44,45]. Since there is no significant difference between the consensus process of the global variables of ADMM or other types of messages, interested readers are referred to these papers for further reading. We note that network coding can improve both the reliability and security of the consensus, i.e., as secure network codes [46].

## 6. Conclusions and Future Work

We discussed how coding can be used to improve the reliability and reduce the communication loads for both primal- and primal–dual-based DML. We discussed both deterministic (and optimal) and random construction of error-control codes for DML. For the low-complexity and high flexibility, the latter may be more suitable for large-scale DML. For primal-dual based DML (i.e., ADMM), we discussed separate coding process for the two steps of ADMM, i.e., in local optimization and consensus processes separately. We introduced the algorithms on how to use codes for the local optimization of ADMM.

For emerging applications of increased interest, DML will be more and more common. Another interesting area for applying coding for DML is security. Though DML has a certain privacy-preserved capability (compared to transmit raw data), a higher security standard may be needed for sensitive applications. Secure coding has been an active topic for years; see [47]. We also have preliminary results on improving privacy by artificial noise in DML [40]. However, a further study is largely needed for improving performance and general scenarios.

Another interesting area for future work may be further studying coding for primal–dual methods. Though separate coding for the two steps of ADMM may solve the problem partly, the coding efficiency may be low and system complexity may be high. As discussed in Section 5, directly applying error control codes to ADMM may be infeasible. Another potential approach may be to simplify the optimization functions without significant performance loss, and error-control codes can be used.

## Figures and Tables

**Figure 1 entropy-24-01284-f001:**
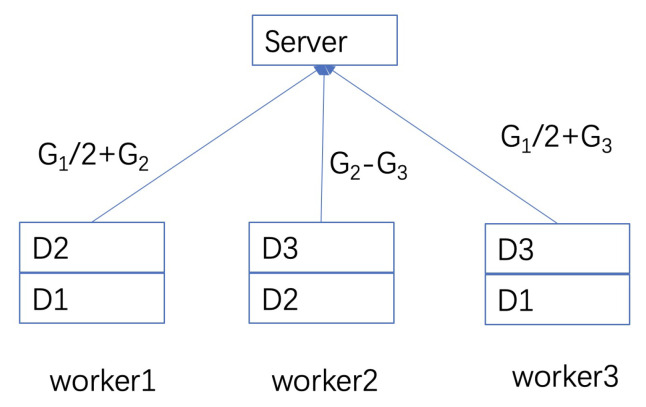
Coded DML with a master–worker structure can tolerate any of one straggler node.

**Figure 2 entropy-24-01284-f002:**
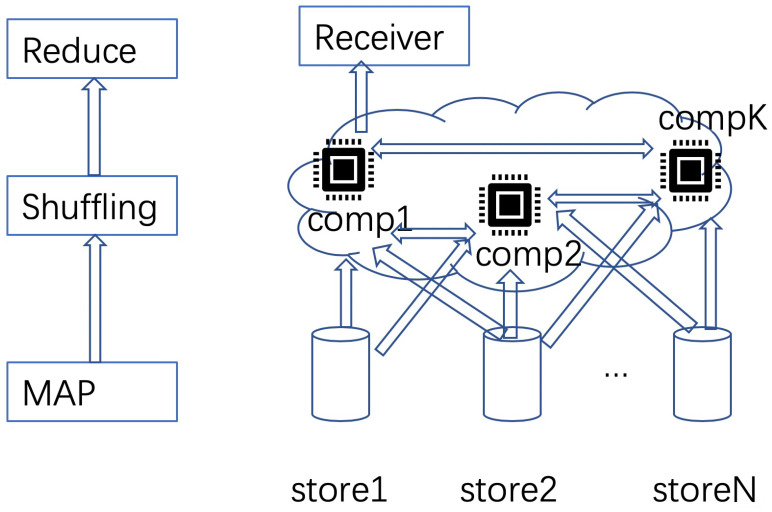
A common realization of DML based on MAPReduce.

**Figure 3 entropy-24-01284-f003:**
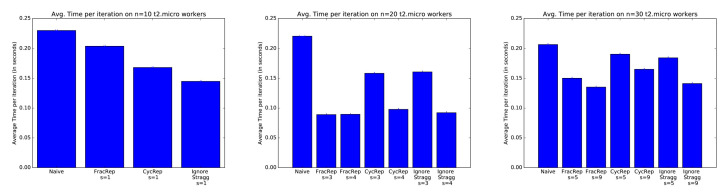
Comparison average time per iteration on Amazon employee access dataset [34].

**Figure 4 entropy-24-01284-f004:**
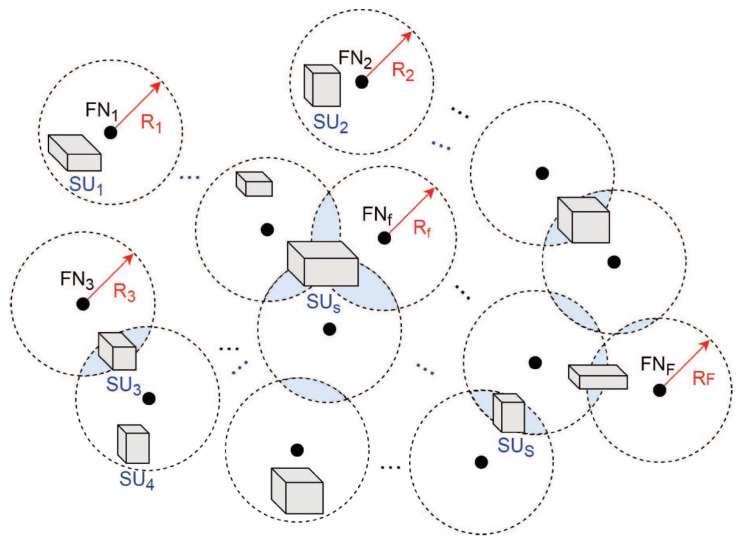
Distributed machine learning with multiple data storage and computing/fog nodes.

**Figure 5 entropy-24-01284-f005:**
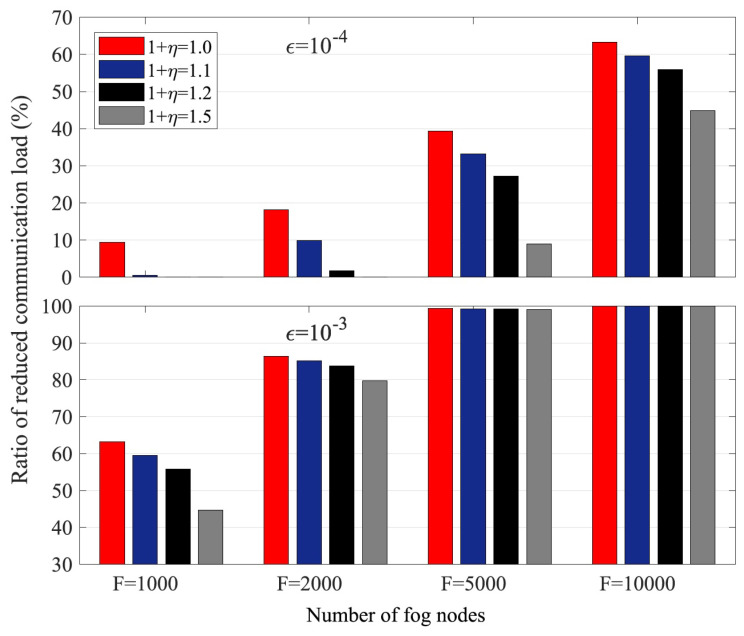
Ratio of coding gains relative to uncoded systems in communication loads.

**Figure 6 entropy-24-01284-f006:**
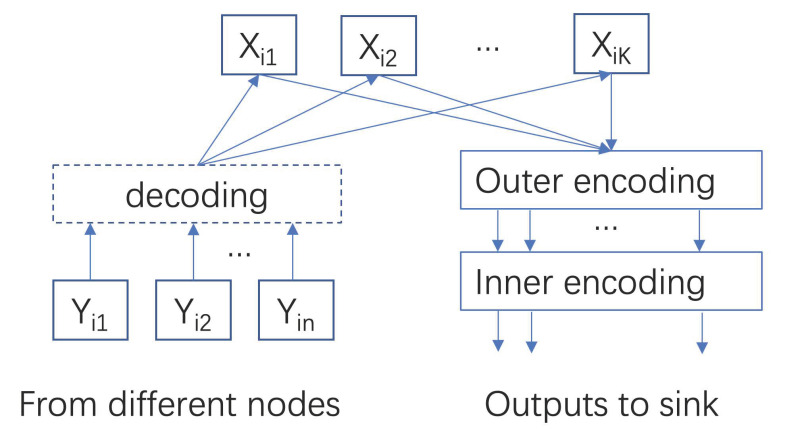
Large-scale distributed machine learning (DML) with BATS codes.

**Figure 7 entropy-24-01284-f007:**
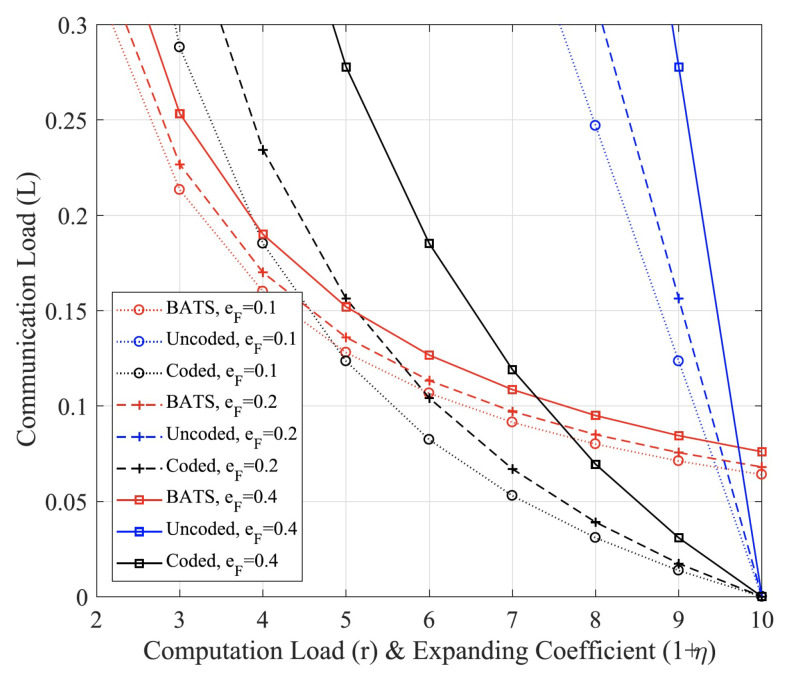
Communication load comparison among BATS codes, coded computing (deterministic codes) and uncoded [12]. $e_{F}$ denotes the channel erasure probability and corresponds to straggling probability. The computing load is defined as involved computing nodes and thus corresponds to expanding coefficients.

**Figure 8 entropy-24-01284-f008:**
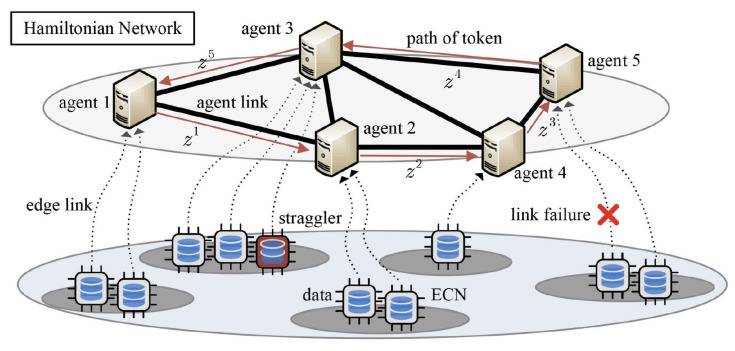
ADMM with multiple agents, each of which collect trained models from multiple ECNs with sensed data. Agents are connected via Hamiltonian networks.

**Figure 9 entropy-24-01284-f009:**
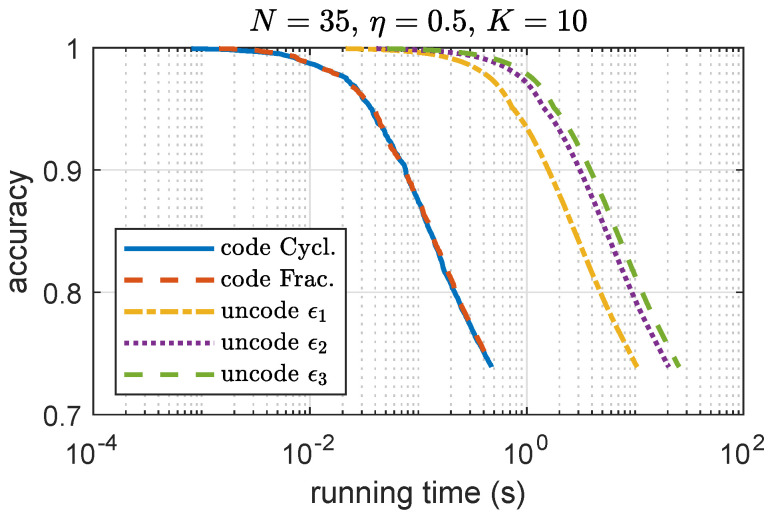
Comparison of coded and uncoded ADMM in accuracy and running time.

**Figure 10 entropy-24-01284-f010:**
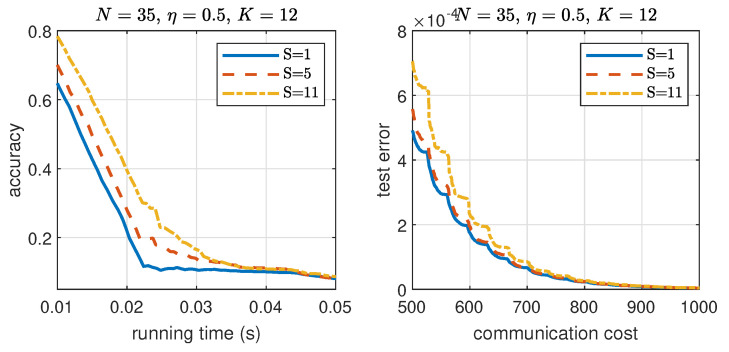
Impact of number of straggler nodes on the convergence rate of the proposed csI-ADMM on synthetic dataset.

## Data Availability

Not applicable.

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
