# Peer review of "Coding for Large-Scale Distributed Machine Learning"

_entropy, 2022, doi:10.3390/e24091284_

Round 1

Reviewer 1 Report

The submitted paper presents a comprehensive review of coding for large-scale distributed machine learning (DML). After a brief introduction of DML in Sec. 2, Sec. 3 reviews gradient coding [34]. Sec. 4 reviews authors' own coding schemes [13, 14] based on Fountain codes. Sec. 5 reviews coding [15] for alternating direction method of multipliers (ADMM) before conclusions in Sec. 6.

The submitted paper provides a comprehensive review of importants issues: coding in DML. The main target of the paper may be coding theorists, who are the main readers of Entropy. However, the presention quality of Sec. 3 may be a little low. Thus, the paper should be published after minor revisions. 

Minor comments: 

--In page 5, 

Clarify the definitions of the two matrices A and B, such as the dimensions and its meaning of A and B, the meaning of rows, and etc. In my understanding, A and B are f1 \times f2 and f2 \times k, respectively. f1, f2, and k denote the number of "correctable" straggler paterns, the number of workers, and the number of gradients. The paper assumes f2=k=3 implicitly. (Assume it explicitly.) As a result, f1=3 holds. 

B is an encoding matrix, i.e. Bg represents a message vector workers send to the server.  A is a "decoding" matrix satisfying ABg=(1^{T}g)1. Each row of A represents how to combine the message vector for each straggler pattern (The ith row means worker i is a straggler).  Present the example (3) after such clear definitions of A and B. 

--page 6, lines 226 and 227

Clarify the definition of the second step in the fractionl repetition construction. In my understanding, the precise definition is presented in lines 232--235, "In a group, the first worker ... and so on."  Present the definition before Eq. (4), in which there might be typos, i.e. should Eq. (4) be block diagonal?  

--Condition 1 

Clarify what is the condition. In my understandng, "for every subset I\subset span{...}" is a condition. My recommendation is "Consider any scheme ... Then, every subset I\subset  span{...} is satisfied." At least, separate the single sentence into the assumption and conclusion sentences. 

--Ignoring straggler scheme in numerical results

Clarify the meaning of the ignoring straggler scheme.  Fig. 3 implies that there are no points in using CycRep because CycRep is inferior to the ignoring straggler scheme. Or is the ignoring straggler scheme kind of lower bounds? 

--Fig. 9

Clarify and fix the legends in Fig. 9. The third and last are identical. What are \epsilon_1 and \epsilon_2?

Author Response

Comments:

  1. The submitted paper presents a comprehensive review of coding for large-scale distributed machine learning (DML). After a brief introduction of DML in Sec. 2, Sec. 3 reviews gradient coding [34]. Sec. 4 reviews authors' own coding schemes [13, 14] based on Fountain codes. Sec. 5 reviews coding [15] for alternating direction method of multipliers (ADMM) before conclusions in Sec. 6.

The submitted paper provides a comprehensive review of importants issues: coding in DML. The main target of the paper may be coding theorists, who are the main readers of Entropy. However, the presention quality of Sec. 3 may be a little low. Thus, the paper should be published after minor revisions. 

Minor comments: 

--In page 5, Clarify the definitions of the two matrices A and B, such as the dimensions and its meaning of A and B, the meaning of rows, and etc. In my understanding, A and B are f1 \times f2 and f2 \times k, respectively. f1, f2, and k denote the number of "correctable" straggler paterns, the number of workers, and the number of gradients. The paper assumes f2=k=3 implicitly. (Assume it explicitly.) As a result, f1=3 holds. B is an encoding matrix, i.e. Bg represents a message vector workers send to the server.  A is a "decoding" matrix satisfying ABg=(1^{T}g)1. Each row of A represents how to combine the message vector for each straggler pattern (The ith row means worker i is a straggler).  Present the example (3) after such clear definitions of A and B. 

R1: Thanks for your comments. Following your comments, we have revised the paper in accordingly Page 5 and 6. We first introduced the definition of A and B and then gave the example.

  1. --page 6, lines 226 and 227

Clarify the definition of the second step in the fractionl repetition construction. In my understanding, the precise definition is presented in lines 232--235, "In a group, the first worker ... and so on."  Present the definition before Eq. (4), in which there might be typos, i.e. should Eq. (4) be block diagonal?  

R2: Thanks for your comments. We have revised the paper accordingly. Yes, you are right. (4) has typo and it should be a block diagonal matrix.

  1. --Condition 1 

Clarify what is the condition. In my understandng, "for every subset I\subset span{...}" is a condition. My recommendation is "Consider any scheme ... Then, every subset I\subset  span{...} is satisfied." At least, separate the single sentence into the assumption and conclusion sentences. 

R3: Thanks for the comments. Following your comment, we have revised Condition 1 accordingly.

  1. --Ignoring straggler scheme in numerical results. Clarify the meaning of the ignoring straggler scheme.  Fig. 3 implies that there are no points in using CycRep because CycRep is inferior to the ignoring straggler scheme. Or is the ignoring straggler scheme kind of lower bounds? 

R4: Thanks for the comment. Following your comment, we have added the explanation of Ignoring straggler scheme in the end of Page 7. Note ignoring straggler node may indeed be fast. However, it may lose performance in generalization errors, since it may lose model parameters (lose data) by simply ignoring stragglers. We also note that global iteration can still proceed with smaller data set in ignoring straggler scheme.

  1. --Fig. 9 Clarify and fix the legends in Fig. 9. The third and last are identical. What are \epsilon_1 and \epsilon_2?

R5. Thanks for your comments. Yes, there is typo in Fig. 9. There is a \epsilon_3. Following your comments, we update the figure and also explain the meaning of \epsilon in the end of Page 15.

Here \epsilon denotes the maximum tolerate time for a straggler node (maximum wait time). We assume \epsilon_1 > \epsilon_2 > \epsilon_3.

Reviewer 2 Report

This paper provides an overview of the coding for large-scale distributed machine learning, with emphasis on the introduction to the random coding for gradient based DML and the separate coding processes for primal-dual based DML (ADMM).  The paper is well written, and the overview is comprehensive and insightful. I think it merits to be published.

I have the following comments for minor revision of the manuscript.

1) Some phrases are not consistent throughout the paper. For instance, MAPReduce is written as MAPReduce on page 5 and MapReduce on page 12.

2) Page 4, Figure 1: Why does a worker node have two different data parts (such as D_1 and D_2) instead of one (such as D_1)? It would be better to give an brief explanation.

3) The notation of all 1s matrix in “AB=1” on pp.5 and the “1” in equ. (4) are different.

4) pp.13, lines 4-5, “and then coded version of coded version”. Does it mean “coded version of sI-ADMM” ?

5) In Algorithm 4 on pp.15, “divide D_i labeled data based on repetition schemes in [? ]”. Lack of reference number.

Author Response

Thanks for the reviewer for valuable comments for our manuscripts, which help us to improve the paper. The one-by-one response is as follows. The revisions in the paper are highlighted in bold text.

  1. Thanks for pointing out the inconsistence of our paper. Following your comments, we have revised the paper. We only keep “MAPReduce” throughout the paper.
  2. Thanks for your comments. The data set of a worker node is divided into different sets are due to following reasons: (1), Different nodes may have partly overlapped data sets in practice. Thus, it is efficient to divide the data set of a work node into small sets and then two nodes may have a few identical sets. (2), Coding of DML is in different nodes and there is no communication among them. Thus, it is necessary that different nodes have partly the same data sets to enable coding. (3), In terms of coding, each data set (or related gradient) can be regarded as a code symbol. Many code symbols are often necessary for encoding. Following your comments, we have revised the paper in Page 4, Figure 1 as follows:

As in Figure 1, we divide the data set of a node into multiple smaller sets to denote the partial overlapping of different nodes. Meanwhile, multiple sets in a node are also necessary for encoding as shown in the figure since one data set corresponds to one source symbol of the code.

  1. Thanks for your comment. We have revised the notation in pp. 5 for consistence.
  2. You are right. It is a typo. We have revised it accordingly.
  3. Thanks for pointing out the typo. We have revised the paper accordingly.